# Hollow-Structured Microporous Organic Networks Adsorbents Enabled Specific and Sensitive Identification and Determination of Aflatoxins

**DOI:** 10.3390/toxins14020137

**Published:** 2022-02-13

**Authors:** Lu Yang, Jin Wang, Huan Lv, Xue-Meng Ji, Jing-Min Liu, Shuo Wang

**Affiliations:** Tianjin Key Laboratory of Food Science and Health, School of Medicine, Nankai University, Tianjin 300071, China; yl1215924627@163.com (L.Y.); wangjin@nankai.edu.cn (J.W.); lvhuan@nankai.edu.cn (H.L.); jixuemeng@nankai.edu.cn (X.-M.J.)

**Keywords:** food-safety inspection, solid-phase extraction, aflatoxins, HMONs, adsorbents

## Abstract

Aflatoxin (AFT) contamination, commonly in foods and grains with extremely low content while high toxicity, has caused serious economic and health problems worldwide. Now researchers are making an effort to develop nanomaterials with remarkable adsorption capacity for the identification, determination and regulation of AFT. Herein, we constructed a novel hollow-structured microporous organic networks (HMONs) material. On the basis of Fe_3_O_4_@MOF@MON, hydrofluoric acid (HF) was introduced to remove the transferable metal organic framework (MOF) to give hollow MON structures. Compared to the original Fe_3_O_4_@MOF@MON, HMON showed improved surface area and typical hollow cavities, thus increasing the adsorption capacity. More importantly, AFT is a hydrophobic substance, and our constructed HMON had a higher water contact angle, greatly enhancing the adsorption affinity. From that, the solid phase extraction (SPE-HPLC) method developed based on HMONs was applied to analyze four kinds of actual samples, with satisfied recoveries of 85–98%. This work provided a specific and sensitive method for the identification and determination of AFT in the food matrix and demonstrated the great potential of HMONs in the field of the identification and control of mycotoxins.

## 1. Introduction

Food is the basis of all human life activities. In recent years, the frequent occurrence of malignant events caused by food safety problems, especially the biotoxin residues in food, resulted in inestimable harm and loss [1]. In order to meet the social demand for food safety risk control, the development and research of fast, sensitive and accurate detection of food hazards has become very urgent. Recently, the major hazards to human health from foods are mycotoxins, drug residues, food pathogens, heavy metal ions, food additives and allergens. Among these, mycotoxins have attracted more and more attention from the public due to their significant toxicity to humans and the fact that they are easily-produced in the food matrix.

Countries around the world now are putting great effort into preventing mycotoxin development in foods to ensure the commercial transit and the free movement of goods because the food-borne pathogens along with the corresponding mycotoxins are of utmost concern in society. The Food and Agriculture Organization of the United Nations (FAO) concluded that over 25% of cereals hold the risk of contamination by mycotoxins over the world [2]. Mycotoxin pollution not only poses a huge threat to the food safety of citizens but is also the biggest obstacle to China’s agricultural products export to the EU, causing huge economic losses to China’s grain and oil processing and export enterprises. Aflatoxin (AFT) is a difuran ring toxoid produced by fungi such as Aspergillus flavus [3]. AFG1, AFG2, AFB1 and AFB2 are the four most basic types with acute and chronic toxicity [4,5]. Grains are susceptible to mycotoxin contamination in the process of growth, harvest, storage and transportation [6]. Among these, corn, rice, soybean and millet are the most common. With the increase of people’s attention to food safety, the problem of aflatoxin contamination is increasingly concerned and valued by governments all over the world, and countries have developed aflatoxin sales standards to protect their food safety and trade interests. The Commission Regulation (EC) 1881/2006 sets limits of 2.0 and 4.0 ppb for aflatoxin B1 and aflatoxin totals for all cereals and all cereal derivatives, respectively [7]. The Food and Drug Administration (FDA) stipulates that 20 ppb of aflatoxin is the upper limit for human food and products [8]. China GB 2761-2017 National Standards for Food Safety Limits of mycotoxins in food also set limits for five mycotoxins in grain and oil foods [9]. Mycotoxin limits in food are established on the basis of risk assessment, based on data from food safety risk monitoring and dietary exposure of residents, as well as factors such as existing regulations of trading partner countries and whether food demand can be met under such regulations. In different countries and regions, due to the differences in climate, environment and food types, the dietary structure and dietary exposure of residents are not the same, so the limits of mycotoxins in food are also different to some extent. In terms of mycotoxin controlling, the EU requires every operator in the food chain to self-test for mycotoxins in the products they sell and acquire, while China organizes national inspection through the grain department in order to investigate the harvest quality of grain. At present, the basic work of risk analysis and hazard control of mycotoxins in China still needs to be further strengthened so as to constantly improve the limit standards, promote the relevant standards to be in line with the international standards and accelerate the healthy development of China’s food import and export trade.

At present, common analytical methods for the determination of AFT are instrumental detection, such as HPLC and LC-MS, an enzyme-linked immunosorbent assay (ELISA)-based immunoassay and the aptamer screening method [10]. Among them, the enzyme-linked immunoassay has strict reaction conditions and is prone to false-positive results due to enzyme instability [11], while the aptamer method is also affected by environmental variables such as salt concentration and pH value due to its high sensitivity. Instrumental detection is a relatively robust and reproducible method. HPLC has the advantages of wide application range, high separation efficiency, high speed, a wide selection of mobile phase, a wide variety of stationary phases, high sensitivity, repeatable chromatographic columns and safety [12,13,14,15,16]. Due to the extremely low-level presence in the food matrix with the significantly high toxicity of AFT, there appears high standard demand for the sensitivity and anti-interference ability of the developed methods [17,18]. Therefore, appropriate sample pretreatment procedures (enrichment or adsorption) combined with HPLC are thought to be a promising choice for AFT determination in complex food matrices (electrochemically active components, such as catechins, gallic acid, vitamins and inactive components such as proteins, lipids, polysaccharides).

In recent years, with the rapid development of science and technology, functional micro-/nano-materials have gradually entered people’s vision, especially in the field of food safety [19,20,21,22,23,24]. Some porous morphologies, such as MOFs and microporous organic networks (MONs), are widely utilized in food sample pretreatment. As conventional porous-structured materials, MOF materials are 3D crystalline micro-materials with a high surface area and tunable structure; however, their relatively poor chemical stability limits their intensive application in food hazards adsorption [22]. Hybrid porous structures usually provide improved enrichment performance. A novel type of hybrid materials, Fe_3_O_4_@MOF@MON, was reported by Li et al. in 2019 [23]. It exhibited good magnetic separation ability, which effectively simplified pretreatment steps and could be applied to determine the trace AFT in real food samples. In comparison, MONs, a subclass of conjugated microporous polymers (CMPs), have much stronger chemical stability than MOFs [25,26,27,28]. MONs are emerging porous materials consisting of aromatic alkynes and halides via Sonogashira–Hagihara coupling [29]. The prepared MONs have high pore intensity, but this mostly comes from the MONs surface pores, while the majority of interior pores fail to function due to inaccessibility. In fact, this is a common problem faced by MONs in many applications. With the gradual understanding of the growth mechanism and microstructure of nanocrystals by researchers, hollow nanomaterials have become one of the hot spots in modern nanoscience research [30,31,32]. To enhance the adsorption capacity while increasing the utilization of interior pores, current research has become increasingly focused on HMONs. The existence of macropores can reduce the diffusion resistance of analytes in HMONs, shorten the diffusion pathway and increase the mass transfer rate so that analytes molecules can easily access the interior [33,34,35,36]. Currently, HMONs have a wide range of applications in the field of drug release, and further development is needed for research in the field of food safety detection [37,38,39,40,41,42].

In this work, a simple and new method was introduced for HMONs preparation using HF to remove the transferable MOF (green layer) and Fe_3_O_4_ (purple ball) core from Fe_3_O_4_@MOF@MON, as shown in Figure 1. Due to the low content of AFT in actual samples and large matrix interference, enrichment and extraction is a crucial step. Compared to the original Fe_3_O_4_@MOF@MON, HMON gave improved surface area and typical hollow cavities, thus increasing adsorption capacity. More importantly, AFT is a hydrophobic substance, and our constructed HMON had a higher water contact angle, greatly enhancing the adsorption affinity. The SPE-HPLC developed based on HMONs was applied to the analysis of four kinds of actual samples, with satisfied recoveries of 85–98%. This work provided a specific and sensitive method to AFT identification, determination and control in the food matrix and demonstrated that HMONs have great potential in the field of identification and control.

## 2. Results and Discussion

### 2.1. Synthesis and Characterization of HMONs

In this work, a new HMONs structure was prepared using HF to etch Fe_3_O_4_@MOF from Fe_3_O_4_@MOF@MON. (See detailed experimental procedures in the Appendix A) The large specific surface area and strong hydrophobicity made HMONs more qualified for the detection of AFT compared with the original Fe_3_O_4_@MOF@MON. Chemical changes of the surface properties of HMONs in comparison with the original Fe_3_O_4_@MOF@MON were explored by water contact angle measurement. Different thicknesses of HMONs produced different adsorption capacities and surface hydrophobicity. The thickness of HMONs-1, HMONs-2, HMONs-3 and HMONs-4 increased in the order of 6.5, 11.4, 18.9 and 22.3 nm, respectively. With the change of material thickness, there was no significant difference in water contact angle; all had strong hydrophobicity. As shown in Figure 2, the water contact angles gradually increased from 132.1° (Fe_3_O_4_@MOF@MON) to 145.3° (HMONs-1) to 147.5° (HMONs-2), up to 148.5° (HMONs-4). From here, we see that the prepared HMONs had stronger hydrophobicity. For the hydrophobic AFT, the HMONs were ideal adsorbents. Meanwhile, as the thickness of the material increased, there was a significant decrease in adsorption capacity. As we can see, Fe_3_O_4_@MOF@MON showed poor adsorption (21.3 mg g^−1^) for AFT. In comparison, HMONs-1 showed better adsorption. The amount of AFT absorbed decreased gradually from HMONs-1 (89.4 mg g^−1^) to HMONs-4 (39.2 mg g^−1^). This indicated the hollow structure of HMONs possessed a larger specific surface area and higher adsorption capacity than Fe_3_O_4_@MOF@MON. In the meantime, the thinner shell of hollow MONs would lead to better adsorption capacity. Although the HMONs-1 had the best adsorption capacity and stronger hydrophobicity, we chose the HMONs-2 as a follow-up experimental material. That is because the HMONs-1 thickness was too thin, resulting in poor stability; besides, it had lower density, which required high centrifugation speed during analysis. All things considered, HMONs-2 were chosen as the final materials used for the followed SPE-HPLC assay.

As shown in Figure 3, the typical transmission electron microscopy (TEM) images clearly showed the shape of Fe_3_O_4_@MOF@MON (Figure 3a–c) as well as the hollow structured HMONs (Figure 3d–f). According to TEM studies, it was seen that both of them showed a glomerated sphere form and average particle sizes approaching 200 nm, which is similar to the average hydrodynamic particle size obtained by dynamic light scattering (Figure 3g–h). At least 150 randomly selected particles were measured from the TEM images, the average diameter of Fe_3_O_4_@MOF@MON and HMONs was calculated as 198.9 ± 7.4 nm and 185.4 ± 9.7 nm, respectively. The above results showed, in comparison with Fe_3_O_4_@MOF@MON, the particle size of HMON encountered no significant change as well as the uniformity of particles.

For the sake of confirming the large specific surface area and hollow structure of HMONs, the specific surface area of BET (Brunauer–Emmett–Teller) and the pore size distribution of BJH (Barrett–Joiner–Halenda) were analyzed by nitrogen adsorption–desorption isotherm. Figure 4A,B show the N_2_ adsorption–desorption curve of Fe_3_O_4_@MOF@MON and HMONs. This type of desorption–adsorption curve belongs to type II in the IUPAC (International Union of Pure and Applied Chemistry) classification. Type II isotherms reflect typical physical adsorption processes on microporous adsorbents. Figure 4C show that the BET surface area of Fe_3_O_4_@MOF@MON was only 468.7 m^2^ g^−1^, while HMONs increased to 897.3 m^2^ g^−1^. Generally speaking, a high specific surface area could offer more active sites and thereby improve adsorption efficiency. The inset of Figure 4F show that the total pore volume of Fe_3_O_4_@MOF@MON and HMONs is about 1.25 mL g^−1^ and 1.78 mL g^−1^, respectively. The results revealed that this kind of hollow material with high specific surface area and high pore volume is very promising in its application to food-borne hazard determination.

Besides high specific surface area, HMONs also showed excellent chemical stability. Since the physicochemical properties of nanoscale materials depend on size, instability refers to the increase or decrease in nanoparticle size that leads to significant changes in the catalytic, optical, magnetic, mechanical, and thermal properties of the material. We characterized the particle size and surface area change at pH 4, 7 and 10 for 48 h. According to Figure 4D,E, HMON particle size and the surface area changed little, so it can be seen that HMONs were very stable under different pH environments, which ensures the developed HMONs materials could be applied for adsorption in various pH environments. Chemical stability is the basis of HMON’s practical application in food substrate interference.

All these results proved that HMONs had higher adsorption capacity and stronger hydrophobicity than Fe_3_O_4_@MOF@MON and was more suitable as an adsorbent for AFT.

### 2.2. Adsorption Performance

The adsorption performance of HMONs for AFT was characterized by equilibrium experiments and adsorption kinetics. (See detailed experimental procedures in the Appendix A) As we can see from Figure 5A, the adsorption capacity increased with the concentration in the range of 0–30 mg L^−1^, almost reaching balance at the concentration of 20 mg L^−1^. Meanwhile, the adsorption capacity of HMONs increased significantly in the first five minutes and almost reached the peak after 10 min. The adsorption capacity of HMONs to AFT was 82.3 mg g^−1^ (Figure 5B). These results indicated that as-prepared HMONs possessed the excellent advantages of efficient and fast adsorption of AFT. Initial experiments were related to the comparison of the adsorption features of prepared nanocomposite and its components. Both adsorbents were applied to the extraction of AFTs from their solutions (20 mg L^−1^). After the desorption step in acetonitrile, the amounts of extracted AFTs were evaluated. As we can see, the adsorption efficiencies of HMONs (76.3, 67.1, 72.3, 81.8 mg g^−1^ for AFB1, AFB2, AFG1, AFG2, respectively) were greatly equivalent to about four times those of the pristine Fe_3_O_4_@MOF@MON (18.8, 17.7, 16.4, 17.4 mg g^−1^ for AFB1, AFB2, AFG1, AFG2, respectively). This is because the hollow structure confirmed HMONs’ large specific surface area and more binding cavities to AFTs. In general, a higher specific surface area indicated more active sites.

### 2.3. Optimization of SPE-HPLC Conditions

As for the SPE-HPLC optimization experiments, several key factors, including the adsorbent amount, sampling pH, elution volume, and eluent composition, on the adsorption capacity of HMONs were evaluated using an AFT standard solution of 100 μg L^−1^. The adsorbent amount used for the SPE assay was evaluated with an amount ranging from 10 to 50 mg. Results in Figure 6A demonstrated that increasing adsorbent amounts gave similar recovery performance, revealing that 10 mg is a sufficient amount for this assay. Moreover, 10 mg of adsorbents were applied for the following optimization experiments.

As the pKa of AFT is 10.09 ± 0.20, the AFT in the grain sample remained neutral at pH 10. The recoveries gradually decreased when pH > 9.0, possibly because the deprotonation would weaken the interaction between AFT and the HMONs adsorbent, causing the efficiency of adsorption to be reduced. In addition, AFTs decompose in strongly acidic conditions at a pH lower than 3. The sample pH is a key factor affecting the AFT adsorption process on an adsorbent. In consequence, the influence of sample pH on the adsorption efficiency of the HMONs adsorbent was examined by sample pH from 3.0 to 9.0. Figure 6B show the highest recovery was obtained at pH 6. The extraction efficiency of AFB1, AFB2, AFG1 and AFG2 was increased when the pH value was less than 6.0 but decreased when it was above 6.0.

The type of desorption solvent is also a crucial factor in achieving ideal recoveries. The influence of the type of elution solvent with different polarities was evaluated using acetonitrile, methanol, methanol + formic acid (FA) and acetonitrile + FA. Acetonitrile was selected as the elution solvent since it offered higher extraction recoveries than others (Figure 6D). Consequently, acetonitrile was employed to elute AFT from the adsorbent, and the optimal volume of acetonitrile was then determined within the range of 1.0–10.0 mL. Figure 6C show that as the volume of acetonitrile increased from 1 to 6 mL, the extraction recovery increased gradually. When the volume of elution liquid exceeded 6 mL, the recovery rate did not change, indicating that 6.0 mL of methanol was sufficient for the elution of all AFT from the HMONs adsorbent.

### 2.4. Method Validation

To evaluate the specificity of HMON materials to aflatoxins, several other mycotoxins (sterigmatocystin (ST), Ochratoxin A (OTA), fumonisins, and patulin) are introduced as co-interference. As shown in Figure 7, the presence of the four other mycotoxins gave little effect to the adsorption capacity to HMONs for the four aflatoxins, possibly due to the co-interaction of the HMON hole with the molecular structure of aflatoxins, along with the hydrophobic effect. A series of standard analyte solutions with a concentration range of 0.1–100 μg L^−1^ were prepared to evaluate the practicability of the developed HMONs-SPE-HPLC method for AFT detection. The correlation coefficient (R^2^), linearity, relative standard deviations (RSD) and limits of detection (LODs, S/N = 3) were obtained under optimal conditions (Table 1). The R^2^ values of AFB1, AFB2, AFG1 and AFG2 were all greater than 0.999 in the range of 0.1–100 μg L^−1^ and the LODs were lower than 0.05 μg L^−1^. Compared with Fe_3_O_4_@MOF@MON, the prosed HMONs-based SPE method had a lower LOD, indicating that AFT analysis was applicable to trace level. These results implied that the HMONs adsorbent based on HPLC was a sensitive system for the extraction and detection of trace AFT.

Additionally, to further demonstrate the stability and reusability of the HMONs, they were repeatedly used to adsorb AFT, and the recoveries of AFT were investigated. Figure 8 show that the recoveries of AFTs remained above 90% post 20 cycles of reuse. Meanwhile, previous studies verified that the Fe_3_O_4_@MOF@MON could be reused seven times. Thus, the newly designed HMONs had better stability and applicability for the recovery of AFT, which was greatly in line with the current concept of environmental protection.

### 2.5. Application of Real Samples

After proving HMONs’ strong adsorption capacity and repeatability, we studied its feasibility in real grain samples, including corn, soybean, millet and rice. The recoveries of spiked samples were in the range of 90–98% for corn samples, 88–94% for soybean samples, 85–95% for millet samples and 86–94% for rice samples, respectively (Table 2). The proposed HMONs displayed excellent adsorb performance for AFT in real grain samples. Generally speaking, the developed HMONs-based SPE-HPLC method for detecting AFTs has good linearity, low LOD and a high recovery rate. Compared with Fe_3_O_4_@MOF@MON, HMONs has a larger specific surface area and double hydrophobic properties, which makes them have higher adsorption efficiency and indicates they are more suitable for AFT analysis in actual samples. At last, we summarized the methods of aflatoxin detection based on HPLC in recent years in Table 3. Compared with the existing methods for detecting AFT based on HPLC-SPE, the LODs of this method we reported were lower than average, indicating the HMONs-SPE-HPLC is highly sensitive.

## 3. Conclusions

In conclusion, HMONs are well constructed using HF to remove Fe_3_O_4_@MOF core from Fe_3_O_4_@MOF@MON. A larger surface area was detected for HMONs compared with Fe_3_O_4_@MOF@MON, which increased from 468.7 to 897.3 m^2^ g^−1^. Meanwhile, the water contact angles gradually increased from 132.1° (Fe_3_O_4_@MOF@MON) to 147.5° (HMONs). The strong hydrophobicity improved the adsorptive affinity of HMONs to AFTs. Due to the low content of AFT in actual samples and severe matrix interference, enrichment and extraction are crucial steps. The HMONs were successfully applied as an SPE adsorbent for analysis of AFT contamination in four different grains. The large specific surface and hydrophobicity of both inner and outer bilayers enable HMONs to retrieve aflatoxin from different grain extracts ultra-quickly and efficiently while achieving the “full recovery” of biotoxins with recoveries of 85–98%. In addition, the reactivated HMONs can be reused at least 20 times, which is in line with the current concept of environmental protection. This work showed the specific adsorption capacity of trace aflatoxin in agricultural products, and the hollow structure provided a new idea for the nanomaterials in the field of the identification and control of mycotoxins. Considering the superior properties, hollow structured micro-/nano-adsorbents exhibited considerable potential to detect a variety of hazardous substances in the field of the identification and control of mycotoxins. As for the further practical use of the developed SPE assay for AFT identification and determination in food, future attention should be focused on the scale-up synthesis of HMONs materials and cost-saving to make it affordable to use in practice.

## 4. Materials and Methods

### 4.1. Chemicals and Materials

Zirconium chloride (ZrCl_4_, 98%), *N*,*N*-Dimethylformamide (DMF, >99.9%), 2-aminoterephthalic acid (NH_2_-BDC, >98%), sterigmatocystin (ST, >99.9%), Ochratoxin A (OTA, >99.9%), fumonisins (>99%) and patulin (>99%) were obtained from Aladdin (Shanghai, China). HPLC grade Acetonitrile and Methanol were acquired from Thermo Fisher (Massachusetts, MA, USA). HPLC-grade AFB1, AFB2, AFG1 and AFG2 (HPLC-grade) were purchased from Sigma (Shanghai, China). Deionized water was used in all synthetic experiments. All reagents were at least analytical reagent-grade.

### 4.2. Instrumentation

The water contact angles were measured on a DSA30 contact-angle system (KRÜSS, Berlin, Germany) at room temperature. The N_2_ adsorption–desorption isotherms were characterized to explore the surface area and porous nature using a surface area analyzer (ASAP 2460). The microstructure and the morphology of as-prepared materials were assessed via a transmission electron microscope (TEM, JEM1200EX, Tokyo, Japan). The contents analysis of AFT was measured with high-performance liquid chromatography (HPLC, Shimadzu LC-20AT, Tokyo, Japan).

### 4.3. Synthesis of HMONs

The Fe_3_O_4_@MOF@MON were synthesized as reported previously (See detailed experimental procedures in the Appendix A) [23]. Briefly, Fe_3_O_4_@SiO_2_ (150 mg), Zirconium (IV) chloride (300 mg, 1287 μmol) and water (75 μL) were dissolved in 20 mL N, *N*-dimethylformamide (DMF) and stirred. A solution of 2-aminoterephthalic acid (235 mg, 1298 μmol) in 10 mL DMF was added to the above solution. Then, the mixture was transferred into Teflon-lined autoclaves and heated at 120 °C for 24 h. After cooling the system at room temperature, the solution was thoroughly washed with water five times and then dried under vacuum. Then, the Fe_3_O_4_@SiO_2_@UiO-66-NH_2_ was obtained.

As for the preparation of HMON-2 used in the final SPE assay, in a 100 mL three-necked flask, the prepared Fe_3_O_4_@MOF (200 mg), CuI (1.0 mg, 5.2 μmol) and (PPh_3_)_2_PdCl_2_ (3.4 mg, 4.8 μmol) were dispersed with triethylamine (15 mL) and toluene (15 mL) under sonication and mechanical stir and 1,4-diiodobenzene (80 mg, 0.24 mmol) and akis (4-ethynylphenyl) methane (50 mg, 0.12 mmol) were added. Subsequently, the system was reacted at 90 °C for 6 h. After cooling at room temperature, the acquired substance was separated using a magnet, ultimately washed with methanol and dichloromethane several times to dispose of the unreacted reactants and then dried in a vacuum to finally produce Fe_3_O_4_@MOF@MON. HMONs were obtained by removing Fe_3_O_4_@MOF core from Fe_3_O_4_@MOF@MON by HF solution (5%) treatment for 15 min, then washing with methanol and water. After stirring for 2 h, the obtained product was centrifuged (12,000 rpm, 15 min). Finally, the prepared HMONs were washed five times and dried in a vacuum.

As for the preparation of HMON-1, HMON-3 and HMON-4, the preparation procedures were exactly the same as that of HMON-2 except for the MON reactant ratio in the step of MON coating onto the Fe_3_O_4_@MOF core. The raw materials were CuI (3.9, 6.24, 7.8 μmol), (PPh_3_)_2_PdCl_2_ (3.6, 5.76, 7.2 μmol), 1,4-diiodobenzene (0.18, 0.288, 0.36 mmol) and akis (4-ethynylphenyl) methane (0.09, 0.144, 0.18 mmol) for HMON-1, HMON-3 and HMON-4, respectively.

### 4.4. Sample Preparation

Four grain samples, including corn, rice, soybean and millet, were purchased from the local market. The samples were ground (<80 mesh) and stored at room temperature for 45 days. A total of 5.0 ± 0.1 g sample powder was dispersed in 25 mL of methanol/water mixture (70:30, *v/v*) for full extraction under shaking for 30 min. As for the spiking-recovery test, the aflatoxins were added into the sample powder in this stage to give the final concentration, as shown in Table 2. After filtration, 15 mL of filtrate was removed and it was diluted before passing through the membrane. All the corn samples were given a pre-column derivatization process with trifluoroacetic acid and *n*-hexane and transferred to HPLC vials. Each injection volume was 20 μL.

### 4.5. HPLC Analysis

The AFTs concentrations were measured by an HPLC system (Shimadzu) equipped with LC-20AT FLD-detection (Excitation wavelength: 360 nm and emission wavelength: 440 nm). A Waters Symmetry-C18 column (4.6 mm × 250 mm, 5 μm) at a column temperature of 30 °C was used for the aflatoxin detection. The optimized mobile phase consisted of acetonitrile and water (32:68, *v/v*). The flow rate was 1 mL min^−1^, and the injection volume was 20 μL.

### 4.6. Statistical Analysis

The mean ± SD were determined for all the measured values. Statistical analysis was performed by Student’s *t*-test. The difference between two groups was considered statistically significant for * *p* < 0.1, very significant for ** *p* < 0.01 and the most significant for *** *p* < 0.001.

## Figures and Tables

**Figure 1 toxins-14-00137-f001:**
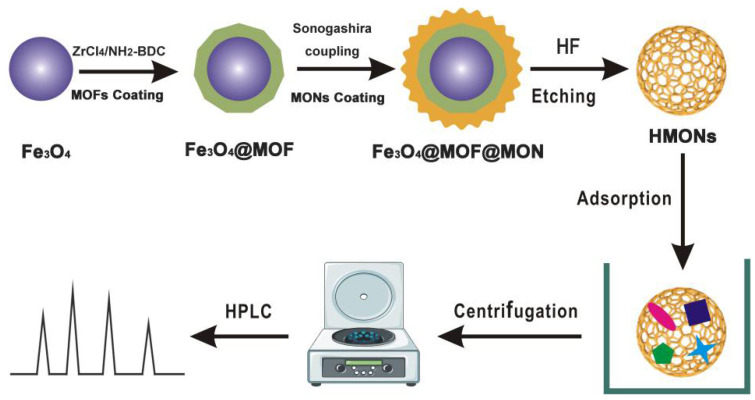
Schematic illustration of the construction of the hollow structured HMONs materials for SPE-based enrichment and determination of aflatoxins in foods.

**Figure 2 toxins-14-00137-f002:**
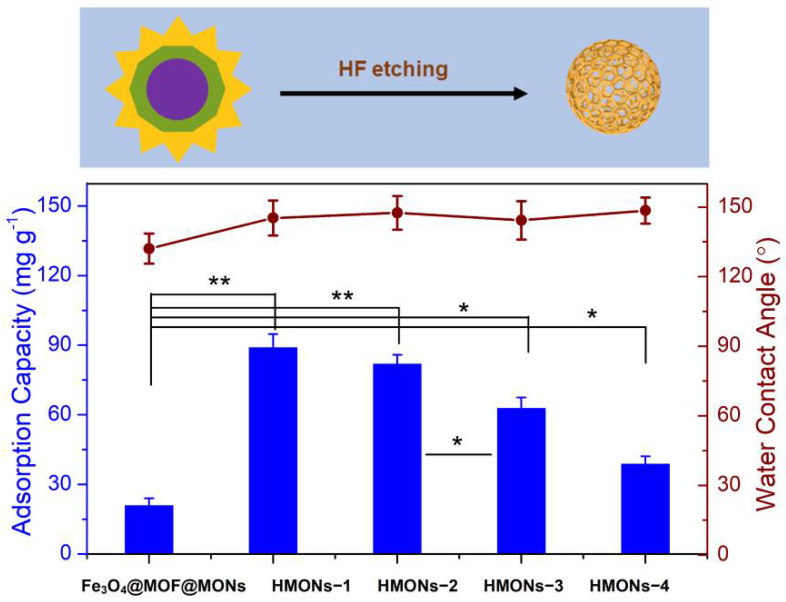
The preparation of hollow structured MONs adsorbents via acid etching, and the comparison of HMONs adsorptive performance with different thicknesses in terms of adsorption capacity (the average value of those to the four targeted aflatoxins) and surface hydrophobicity. The difference between two groups was considered statistically significant for * *p* < 0.1, very significant for ** *p* < 0.01.

**Figure 3 toxins-14-00137-f003:**
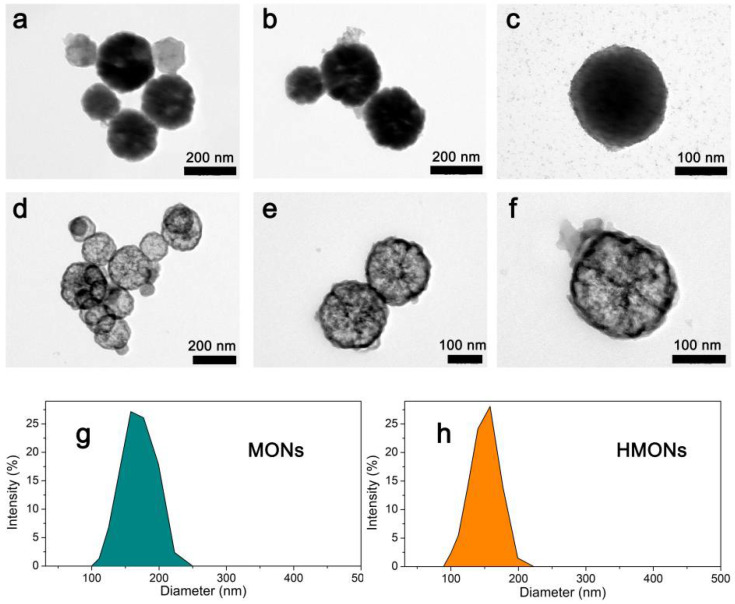
The typical TEM images of the MONs microspheres (**a**–**c**) and HMONs microspheres (**d**–**f**); DLS analysis of MONs (**g**) and HMONs (**h**).

**Figure 4 toxins-14-00137-f004:**
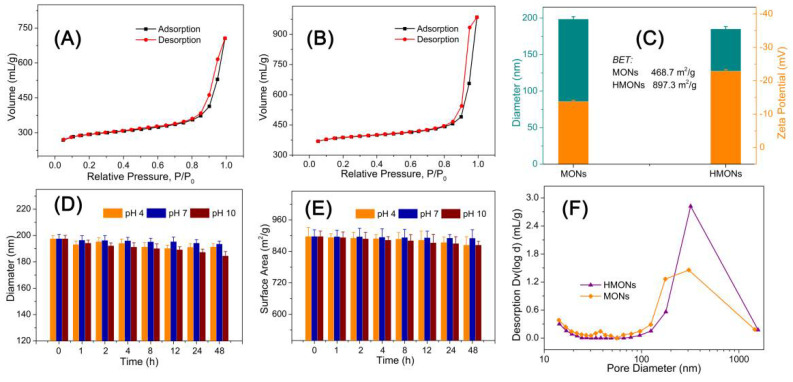
BET analysis of the MONs (**A**) and HMONs (**B**); (**C**) Comparison of the particle size and surface zeta potential of MONs and HMONs; Structural stability evaluation of the developed HMONs microspheres via particle size (**D**) and surface area monitoring at different pH (**E**); (**F**) Comparison of the pore volume of MONs and HMONs.

**Figure 5 toxins-14-00137-f005:**
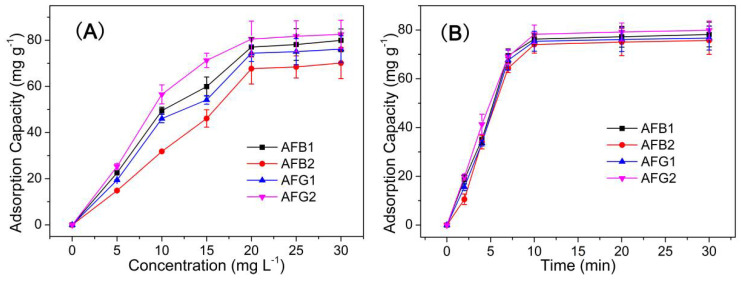
(**A**) Adsorption equilibrium study under an incubated time of 10 min (adsorbents amount, 10 mg; interaction media pH 6; AFT solution of 20 mg L^−^^1^); (**B**) Adsorption kinetic study with the AFT solution of 20 mg L^−^^1^ (adsorbents amount, 10 mg; interaction media pH 6; incubation time 10 min).

**Figure 6 toxins-14-00137-f006:**
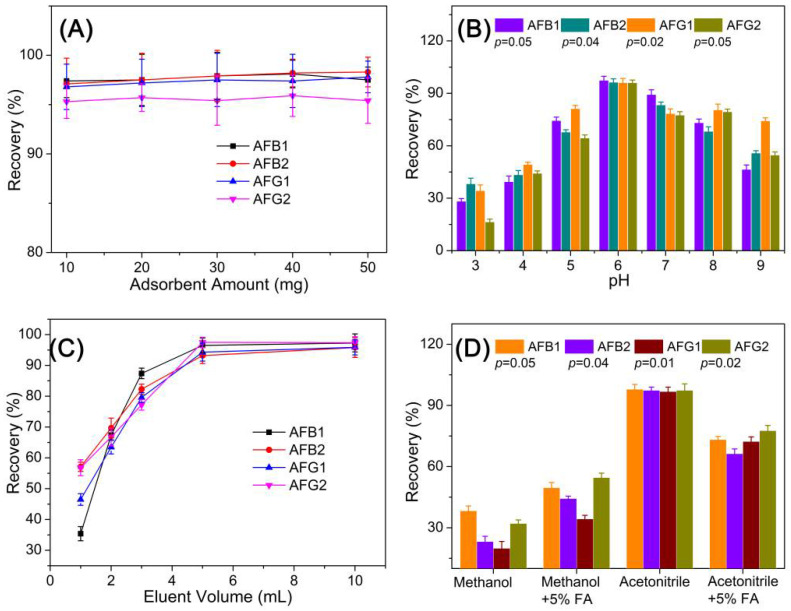
Optimization of the experimental conditions of the developed SPE-HPLC analytical method: (**A**) adsorbent amount; (**B**) incubation pH; (**C**) eluent volume; (**D**) elution solvent.

**Figure 7 toxins-14-00137-f007:**
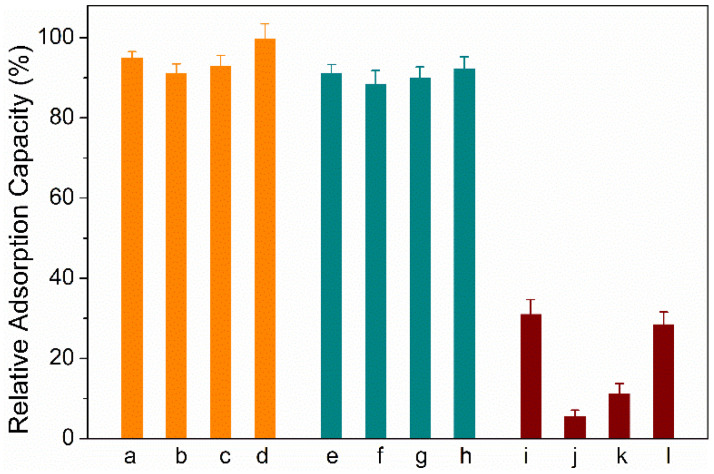
Specificity of the developed HMON adsorption to AFTs: a–d groups represent the AFT B1, B2, G1, G2, respectively; e–h groups represent the AFT B1, B2, G1, G2 in the presence of a mixture of ST, OTA, fumonisins and patulin with the same concentration of AFTs, respectively; i–l groups represent the ST, OTA, fumonisins and patulin, respectively.

**Figure 8 toxins-14-00137-f008:**
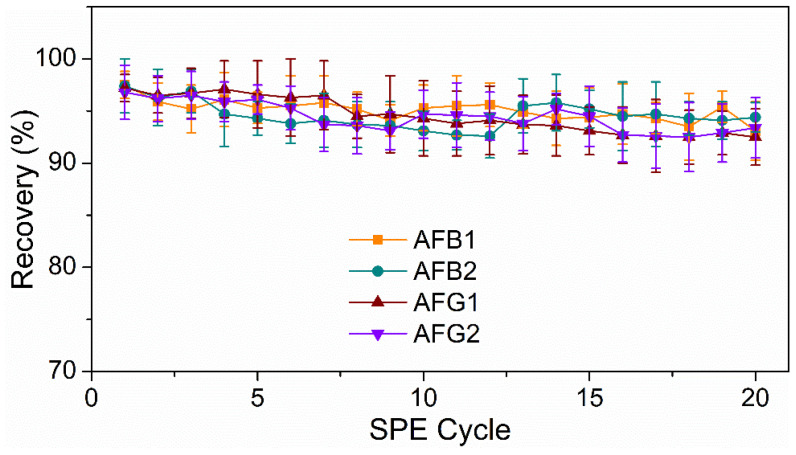
Reusability of the developed HMON-based SPE-HPLC method for AFTs.

**Table 1 toxins-14-00137-t001:** Analytical performance of the proposed HMON-based SPE method for aflatoxins.

Analyte	Retention Time	Linear Range (μg L^−1^)	R^2^	LOD (μg L^−1^)	RSD (%, *n* = 11)
AFB1	4.83	0.1–100	0.9994	0.03	1.9
AFB2	9.76	0.1–100	0.9992	0.04	3.2
AFG1	4.23	0.1–100	0.9993	0.03	2.6
AFG2	7.99	0.1–100	0.9992	0.03	2.8

**Table 2 toxins-14-00137-t002:** Analytical results of real food samples via the developed SPE method.

Analyte		Spiked (μg L^−1^)	Determined Value (mean ± SD, *n* = 3) (μg L^−1^)	Recovery (%)
Corn	AFB1	1	0.93 ± 0.02	93
		10	9.65 ± 0.02	96
	AFB2	1	0.98 ± 0.04	98
		10	9.66 ± 0.06	97
	AFG1	1	0.96 ± 0.06	96
		10	9.71± 0.05	97
	AFG2	1	0.91 ± 0.08	91
		10	9.01 ± 0.02	90
Soybean	AFB1	1	0.89 ± 0.04	89
		10	9.11± 0.05	91
	AFB2	1	0.89 ± 0.05	89
		10	8.98 ± 0.03	90
	AFG1	1	0.88 ± 0.07	88
		10	9.17 ± 0.06	92
	AFG2	1	0.93 ± 0.09	93
		10	9.36 ± 0.04	94
Millet	AFB1	1	0.85 ± 0.05	85
		10	8.78 ± 0.05	88
	AFB2	1	0.92 ± 0.04	92
		10	9.33 ± 0.07	93
	AFG1	1	0.94 ± 0.07	94
		10	9.51 ± 0.05	95
	AFG2	1	0.92 ± 0.08	92
		10	9.48 ± 0.07	95
Rice	AFB1	1	0.91 ± 0.03	91
		10	9.36 ± 0.06	94
	AFB2	1	0.88 ± 0.05	88
		10	9.12 ± 0.04	91
	AFG1	1	0.86 ± 0.02	86
		10	8.97 ± 0.05	90
	AFG2	1	0.87 ± 0.09	87
		10	8.90 ± 0.03	89

**Table 3 toxins-14-00137-t003:** Comparison of existing SPE methods for determination of aflatoxin in food samples.

Sorbent Materials	Detection Method	Analytes	Samples	LODs (μg/L)	Ref
AuNPs@gelatin	HPLC-UV	AFB1	Saffron	0.004	11
AuNBPs@PAF-40-Fe	RP-HPLC-FLD	AFB1	Milk	0.01	9
AuNSs	HPLC	AFB1	Corn flour	0.02	36
Fe_3_O_4_@MOF@MON	HPLC-MSPE	AFT	Corn Rice Millet	0.15	30
MIL53(Al)-SiO_2_@Fe_3_O_4_	HPLC-MSPE	AFB1	Tea	0.5	10
Hydrogel	HPLC	AFB1	Peanut	0.94	8
HMONs	HPLC-SPE	AFT	Corn Rice Millet Soybean	0.03	This method

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
