# Peer review of "Hollow-Structured Microporous Organic Networks Adsorbents Enabled Specific and Sensitive Identification and Determination of Aflatoxins"

_toxins, 2022, doi:10.3390/toxins14020137_

Round 1
Reviewer 1 Report
The previously proposed suggestion (ms ID toxins-1528183) has been only partially developed, since the Authors have cited the legislation applicable in China and the United States, but it is necessary to consider that the Commission Regulation (EC) 1881/2006 sets the maximum levels for certain contaminants in foodstuffs (mycotoxins included), which are lower than those mentioned in the article (eg 20 ppb in the USA), and this aspect is worthy of attention, considering, as you have stated, that "Mycotoxin pollution not only poses a huge threat to the food safety of citizens, but also come to be the biggest obstacle to China's agricultural products export to the EU, causing huge economic losses to China's grain and oil processing and export enterprises". I therefore reiterate the suggestion, asking for a further effort on the examination in the European panorama, like the other Countries mentioned, and citation of the aforementioned European Regulation and on the critical quantitative difference in terms of ppb of mycotoxins allowed with respect to the three analyzed Countries (China, USA and Europe), which in an international journal, accessible to specialists from all over the world, is considered a useful and essential overview.
Author Response
Reply to Reviewer 1
Comment 1:The previously proposed suggestion (ms ID toxins-1528183) has been only partially developed, since the Authors have cited the legislation applicable in China and the United States, but it is necessary to consider that the Commission Regulation (EC) 1881/2006 sets the maximum levels for certain contaminants in foodstuffs (mycotoxins included), which are lower than those mentioned in the article (eg 20 ppb in the USA), and this aspect is worthy of attention, considering, as you have stated, that "Mycotoxin pollution not only poses a huge threat to the food safety of citizens, but also come to be the biggest obstacle to China's agricultural products export to the EU, causing huge economic losses to China's grain and oil processing and export enterprises". I therefore reiterate the suggestion, asking for a further effort on the examination in the European panorama, like the other Countries mentioned, and citation of the aforementioned European Regulation and on the critical quantitative difference in terms of ppb of mycotoxins allowed with respect to the three analyzed Countries (China, USA and Europe), which in an international journal, accessible to specialists from all over the world, is considered a useful and essential overview.
Reply to Comment 1:Thanks for your comments and good suggestions! The limits set by The Commission Regulation (EC) 1881/2006 for mycotoxins have been added to the revised manuscript. Meanwhile, we have analyzed the reasons for the difference in the standard of limits in different countries. (See Revised Manuscript, Page 2)
Reviewer 2 Report
- As a major concern author must provide the results for the adsorption of aflatoxins (AFB1, G1, B2, and G2) in the presence of other mycotoxins such as Ochratoxin A, fumonisins, patulin, etc. to determine the specificity of the prepared HMONs and its industrial applicability.
- Line 4 and 5 in the abstract should be re-writing.
- Reconstruct and shift figure 1 in the material and method section (Make a sematic diagram with sufficient details to explain the synthesis of HMONs.
- In line 314 write the conditions used for the centrifugation.
- Provide the details for the statement mentioned in lines 316 to 318.
- In section 3.4 at what stage aflatoxins were spiked and what is the amount please write it properly.
- Concerning figure 2 adsorption capacity was examined using what? Is it is the mixture of AFB1, AFB2, AFG1, and AFG2 or only AFB1, and what amount was used please mention in the text and in the figure legend accurately.
- In figure 5 legend need to write accurately is it 20 mg/L or 30 mg/L?? At what concentration of aflatoxins and time the study was performed need to write in the figure legends to make it more informative. In symbols inside the figures need to correct also AFT1????AFT2?????
- For lines 200 to 206 where is the data????
- For figure 6 b,c,d experiments were conducted using what amount of adsorbent?
- For line 234 it is better to represent the results as a ratio rather than mL. ex. adsorbent and acetonitrile at x ratio is optimum for the complete elation of aflatoxins.
- Address the typographical and grammatical mistakes throughout the manuscript such as “lager” (line no. 342)
- Besides use the same color in the line and bar graphs to represent a particular type of aflatoxins i.e AFB1, AFG1, AFB2, and AFG2. This will be helpful to read and understand the results.
Author Response
Reply to Reviewer 2
Comment 1:As a major concern author must provide the results for the adsorption of aflatoxins (AFB1, G1, B2, and G2) in the presence of other mycotoxins such as Ochratoxin A, fumonisins, patulin, etc. to determine the specificity of the prepared HMONs and its industrial applicability.
Reply to Comment 1:Thanks for your careful comments! The specificity evaluation of HMON to aflatoxins has been done with four other mycotoxins as co-interference, and the corresponding data has been added as Figure 7. (See Revised Manuscript, Page 7 and 8, Figure 7)
Comment 2:Line 4 and 5 in the abstract should be re-writing.
Reply to Comment 2:Thanks for your careful comments! The corresponding description has been revised. (See Revised Manuscript, Page 1)
Comment 3:Reconstruct and shift figure 1 in the material and method section (Make a sematic diagram with sufficient details to explain the synthesis of HMONs.
Reply to Comment 3:Thanks for your careful comments! The figure 1 has been revised and the detailed procedure description has been added in the main text for clear understanding. (See Revised Manuscript, Page 3 and Figure 1)
Comment 4:In line 314 write the conditions used for the centrifugation.
Reply to Comment 4:Thanks for your careful comments! The experimental details have been added. (See Revised Manuscript, Page 11)
Comment 5:Provide the details for the statement mentioned in lines 316 to 318.
Reply to Comment 5:Thanks for your careful comments! The experimental details have been added. (See Revised Manuscript, Page 11)
Comment 6:In section 3.4 at what stage aflatoxins were spiked and what is the amount please write it properly.
Reply to Comment 6:Thanks for your careful comments! The experimental details have been added. (See Revised Manuscript, Page 11)
Comment 7:Concerning figure 2 adsorption capacity was examined using what? Is it is the mixture of AFB1, AFB2, AFG1, and AFG2 or only AFB1, and what amount was used please mention in the text and in the figure legend accurately.
Reply to Comment 7:Thanks for your careful comments! The experimental details have been added. (See Revised Manuscript, Page 4, Figure 2)
Comment 8:In figure 5 legend need to write accurately is it 20 mg/L or 30 mg/L?? At what concentration of aflatoxins and time the study was performed need to write in the figure legends to make it more informative. In symbols inside the figures need to correct also AFT1????AFT2?????
Reply to Comment 8:Thanks for your careful comments! We have revised the figure and captions to give detailed and accurate information for this part. (See Revised Manuscript, Page 6, Figure 5 and caption)
Comment 9:For lines 200 to 206 where is the data????
Reply to Comment 9:Thanks for your careful comments! The corresponding data of adsorption capacities of both HMON and Fe3O4@MOF@MON have been added. (See Revised Manuscript, Page 6)
Comment 10:For figure 6 b,c,d experiments were conducted using what amount of adsorbent?
Reply to Comment 10:Thanks for your careful comments! All the three experiments were conducted using 10 mg of adsorbents. We have made some supplementations. (See Revised Manuscript, Page 6)
Comment 11:For line 234 it is better to represent the results as a ratio rather than mL. ex. adsorbent and acetonitrile at x ratio is optimum for the complete elation of aflatoxins.
Reply to Comment 11:Thanks for your suggestions! We agree with your comments as ratio would be a clear label for comparison, but in this experiment the ratio would be a relative small number and mL would be more informative as consistent with previously-reported works (J. Hazard. Mater. 2020, 384, 121348; J. Hazard. Mater. 2018, 344, 220-229.)
Comment 12:Address the typographical and grammatical mistakes throughout the manuscript such as “lager” (line no. 342)
Reply to Comment 12:Thanks for your careful comments! We have double-checked the typographical and grammatical mistakes. (See Revised Manuscript)
Comment 13:Besides use the same color in the line and bar graphs to represent a particular type of aflatoxins i.e AFB1, AFG1, AFB2, and AFG2. This will be helpful to read and understand the results.
Reply to Comment 13:Thanks for your good suggestion! We have checked the symbols of all the figures to ensure the same color for the line and bar to represent the AFTs as much as possible, especially Figure 5A and 5B with Figure 6A and 6C. We have made some modifications. (See Revised Manuscript)
Round 2
Reviewer 1 Report
Community legislation has been considered, as per previous suggestions.
However, considering that Reference no. [7] is not related to what is indicated in line 53, I would consider it appropriate to add the following citation:
“COMMISSION REGULATION (EC) No 1881/2006 of 19 December 2006 setting maximum levels for certain contaminants in foodstuffs”
Author Response
Thanks for your good suggestion! We have revised the citation 7 as "COMMISSION REGULATION (EC) No 1881/2006 of 19 December 2006 setting maximum levels for certain contaminants in foodstuffs" as you requested. (See Revised Manuscript, reference 7).
This manuscript is a resubmission of an earlier submission. The following is a list of the peer review reports and author responses from that submission.
Round 1
Reviewer 1 Report
General Comments:
The paper is overall strong and interesting. However, the paper lacks in some critically important areas, specifically in statistical reporting and citation of relevant supporting evidence. The conclusions, particularly about validation and optimization of procedures, are presented only qualitatively in the text, without statistical support. If the paper is to be published, substantial enhancement of reporting of statistical significance needs to be made.
While the paper is clearly a technical one, it would benefit markedly from further contextualization of the proposed method in the realm of available testing methods. What are the practical implications that should be considered, based on the results?
The paper does not describe in detail the limitations of the study, nor does it provide any vision for taking the work forward into the next stages. This would be required to strengthen the paper.
Specific Comments:
L26: The quotation “Food safety first” is not described in context, while the message comes through clearly in the latter half of the second. Delete for clarity, or describe the source/context of the quotation.
L28: What is meant by “harmonious society,” and how does food safety “directly” relate to it? This should be supported with evidence-based examples if kept.
L30: A cited reference (or references) is needed to support the bold claim regarding “inestimable harm and loss.”
L39: “contaminated by aflatoxin, producing this type of toxin” - this sentence is redundant and unclear. I believe the authors are describing the phenomenon whereby the produce is infested by the fungus, which then subsequently produced the toxin in conducive (e.g. poor) storage conditions. This should be clarified.
L42: Cite the source of the statistic 31 million tons of losses
L45: Define the acronyms in full form upon first use - especially “ELISA,” of which the full form is used in the very next sentence.
L54-55: cite a source giving evidence of the “high demand” described. Or is this general knowledge? If so, please describe the context behind the claim clearly.
L68: What does MONs stand for? Define in full form upon first use, if applicable
L99: “newly” what? Check grammar, this sentence is unclear.
L101: how did the authors determine that these parameters were the causal factors leading to being better “qualified,” and what does “qualification” mean in this context? Some elaboration on what indicators were examined is necessary.
L105: A paragraph giving an overview of the HMONs-1 thru HMONs-4 would be greatly helpful. It is hard to place the findings in context without an overview or table describing the composition and distinctions among the different HMONs
L107: What was the statistical test performed to assess significant differences, and what were the test parameters? A more detailed statistical analysis section is needed in the Methods section
L121: “High centrifugal” what? I think the statement is incomplete. Check grammar.
L122: Again, a summary table describing the properties of each variant would be very helpful for understanding why HMONs-2 was selected versus the other candidates.
L139: What was the statistical test performed, and the corresponding parameters? How did the authors assess significance?
L159: How is “zeta potential change” related to the other characteristics described in the prior sentence? Kindly add more detail to firm up the connection.
L161: “especially at pH 7…” What indicates special performance at this pH in particular? It is not clear from the Figure whether there was a significantly better performance at this pH versus the others. Please include statistical tests and make reference to specific figure panels where needed to orient the reader to the results.
L162: “ensured” is not a word that I think applies here - upto this point in the text, no procedure has been reported that “ensures”. I advise focusing on the “potential” described in the next clause.
L196: “clearly showed no obvious difference” - there is nothing clear or obvious without statistical tests! It is necessary to support these observations with proper statistics, otherwise the validity of the optimization procedures cannot be determined.
L212: Was this difference statistically significant? It looks like it was from the plot, but need to report statistics in text clearly.
L217: Was an actual statistical test performed to make this judgement? If not, kindly use a different word than “significant” and describe clearly the basis upon which the judgement was made.
L236: “not a noticeable decrease” - this needs to be supported by a statistical test; moreover, an elaboration on the practical implications of decreased recovery should be given.
L254: Cite references that give evidence of performance characteristics of the other methods, otherwise no sound comparison can be made.
L297: What is the unit of measurement of the “<80”? 80 what?
L302: I cannot find any information about the AFT spiking protocol, nor why the particular spiked concentration was selected. Please add this information or make it more clearly identifiable.
L327: The Conclusions would benefit from further elaboration on practical implications and visions for operationalization. Could this methods be scaled up? Would it be affordable to use in practice, and for which stakeholders? How does the performance x cost trade-off compare to other methods? Cite relevant literature.
Author Response
Reply to Reviewer 1
Comment 1:L26: The quotation “Food safety first” is not described in context, while the message comes through clearly in the latter half of the second. Delete for clarity, or describe the source/context of the quotation.
Reply to Comment 1:Thanks for your comments! We have deleted the quotation “Food safety first”. (See Revised Manuscript, Page 1)
Comment 2:L28: What is meant by “harmonious society,” and how does food safety “directly” relate to it? This should be supported with evidence-based examples if kept.
Reply to Comment 2:Thanks for your careful comments! To avoid any misunderstanding, we have deleted the quotation “food safety is directly related to human health and safety, and have a great impact on the construction of a harmonious society”. (See Revised Manuscript, Page 1)
Comment 3:L30: A cited reference (or references) is needed to support the bold claim regarding “inestimable harm and loss.
Reply to Comment 3:Thanks for your comments! We have cited a reference to support the claim regarding “inestimable harm and loss”. (See Revised Manuscript, Page 1, Ref 1)
Comment 4:L39: “contaminated by aflatoxin, producing this type of toxin” - this sentence is redundant and unclear. I believe the authors are describing the phenomenon whereby the produce is infested by the fungus, which then subsequently produced the toxin in conducive (e.g. poor) storage conditions. This should be clarified.
Reply to Comment 4:Thanks for your careful comments! We have revised“When the grain is not dried in time or stored improperly, it was often easily contaminated by aflatoxin, producing this type of toxin”to “When the grain is not dried in time or stored improperly, it was often easily contaminated by aspergillus flavusaflatoxin, producing this type of toxin”. (See Revised Manuscript, Page 1)
Comment 5:L42: Cite the source of the statistic 31 million tons of losses
Reply to Comment 5:Thanks for your comments! There is no clear reference to support this number, so the 31 million tons of losses has been deleted. (See Revised Manuscript, Page 1)
Comment 6:L45: Define the acronyms in full form upon first use - especially “ELISA,” of which the full form is used in the very next sentence.
Reply to Comment 6:Thanks for your careful comments! We have added the full form of ELISA (Enzyme Linked Immunosorbent Assay). (See Revised Manuscript, Page 2)
Comment 7:L54-55: cite a source giving evidence of the “high demand” described. Or is this general knowledge? If so, please describe the context behind the claim clearly.
Reply to Comment 7:Thanks for your comments! We added the reference 13-14 to show the evidence of the ”high demand” described.(See Revised Manuscript, Page 2)
Comment 8:L68: What does MONs stand for? Define in full form upon first use, if applicable.
Reply to Comment 8:Thanks for your careful comments! The MONs stand for microporous organic networks, we have added the full form in the work. (See Revised Manuscript, Page 2)
Comment 9:L99: “newly” what? Check grammar, this sentence is unclear.
Reply to Comment 9:Thanks for your comments! We have revised the corresponding description. (See Revised Manuscript, Page 3)
Comment 10:L101: how did the authors determine that these parameters were the causal factors leading to being better “qualified,” and what does “qualification” mean in this context? Some elaboration on what indicators were examined is necessary.
Reply to Comment 10:Thanks for your careful comments! We have mentioned “The existence of macropores can reduce the diffusion resistance of analytes in HMONs, shorten the diffusion pathway, increase the mass transfer rate, so that analytes molecules can easily access interior” in Page 2. Meanwhile, aflatoxin is hydrophobic, which made the hydrophobicity of HMONs with great importance. Therefore, the qualification stands for its large specific surface area and strong hydrophobicity.
Comment 11:L105: A paragraph giving an overview of the HMONs-1 thru HMONs-4 would be greatly helpful. It is hard to place the findings in context without an overview or table describing the composition and distinctions among the different HMONs
Reply to Comment 11:Thanks for your comments! The distinctions among the different HMONs are the thickness of MONs shell. We have specified the experimental details about the preparation procedures and raw material concentrations of different HMONs. (See Revised Manuscript, Page 10)
Comment 12:L107: What was the statistical test performed to assess significant differences, and what were the test parameters? A more detailed statistical analysis section is needed in the Methods section.
Reply to Comment 12:Thanks for your careful comments! We have added the explanation about the statistics analysis in the method section 3.6, and statistic analysis results have been supplemented in Figure 2. (See Revised Manuscript, Page 4 and 10)
Comment 13:L121: “High centrifugal” what? I think the statement is incomplete. Check grammar.
Reply to Comment 13:Thanks for your careful comments! The sentence has been revised as “high centrifugation speed where necessary during analysis.” (See Revised Manuscript, Page 3)
Comment 14:L122: Again, a summary table describing the properties of each variant would be very helpful for understanding why HMONs-2 was selected versus the other candidates.
Reply to Comment 14:Thanks for your careful comments! The figure 2 has described the properties of each variant, and we have explained why HMONs-2 is the best choice in Page 3. We have added the explanation about the statistics analysis in the method section 3.6, and statistic analysis results have been supplemented in Figure 2. (See Revised Manuscript, Page 4 and 10)
Comment 15:L139: What was the statistical test performed, and the corresponding parameters? How did the authors assess significance?
Reply to Comment 15:Thanks for your comments! The data of particle size distribution is a trend that illustration of the size distribution of respective particles, not a specific value, thus cannot be compared by statistical analysis. The size distribution is always demonstrated by such styles and commonly used in the previously-reported references. (Biomaterials 2018, 165, 39-47; ACS Applied Materials & Interfaces 2016, 8, 29939-29949)
Comment 16:L159: How is “zeta potential change” related to the other characteristics described in the prior sentence? Kindly add more detail to firm up the connection.
Reply to Comment 16:Thanks for your careful comments! We have revised the wrong description in the sentence. In fact, it should be “We characterized the particle size and surface area change at pH 4, 7 and 10 for 48 h”. (See Revised Manuscript, Page 5)
Comment 17:L161: “especially at pH 7…” What indicates special performance at this pH in particular? It is not clear from the Figure whether there was a significantly better performance at this pH versus the others. Please include statistical tests and make reference to specific figure panels where needed to orient the reader to the results.
Reply to Comment 17:Thanks for your comments! We have deleted the sentence “Especially at pH 7, the surface area and the pore size demonstrated the best stability performance”. In addition, the figure 4D-E illustrated that the HMONs is applicable under various pH environment. (See Revised Manuscript, Page 5)
Comment 18:L162: “ensured” is not a word that I think applies here - upto this point in the text, no procedure has been reported that “ensures”. I advise focusing on the “potential” described in the next clause.
Reply to Comment 18:Thanks for your careful comments! We have revised “The chemical stability ensured the HMONs with great potential for practical applications against food matrix interference” to “Chemical stability is the basis of HMON's practical application in food substrate interference”.(See Revised Manuscript, Page 5)
Comment 19:L196: “clearly showed no obvious difference” - there is nothing clear or obvious without statistical tests! It is necessary to support these observations with proper statistics, otherwise the validity of the optimization procedures cannot be determined.
Reply to Comment 19:Thanks for your careful comments! The sentence “clearly showed no obvious difference” has been revised. (See Revised Manuscript, Page 6)
Comment 20:L212: Was this difference statistically significant? It looks like it was from the plot, but need to report statistics in text clearly.
Reply to Comment 20:Thanks for your comments! We have added the explanation about the statistics in the method section 3.6, and added the statistic analysis results in Figure 6. Some description has been revised to make the conclusion more solid. (See Revised Manuscript, Page 7 and 10)
Comment 21:L217: Was an actual statistical test performed to make this judgement? If not, kindly use a different word than “significant” and describe clearly the basis upon which the judgement was made.
Reply to Comment 21:Thanks for your careful comments! The word “significant” has been deleted. We have added the explanation about the statistics in the method section 3.6, and added the statistic analysis results in Figure 6. Some description has been revised to make the conclusion more solid. (See Revised Manuscript, Page 7 and 10) (See Revised Manuscript, Page 6)
Comment 22:L236: “not a noticeable decrease” - this needs to be supported by a statistical test; moreover, an elaboration on the practical implications of decreased recovery should be given.
Reply to Comment 22:Thanks for your careful comments! The corresponding description has been revised. (See Revised Manuscript, Page 7)
Comment 23:L254: Cite references that give evidence of performance characteristics of the other methods, otherwise no sound comparison can be made.
Reply to Comment 23:Thanks for your careful comments! We have made the comparison in table3.
Comment 24:L297: What is the unit of measurement of the “<80”? 80 what?
Reply to Comment 24:Thanks for your comments! 80 is the mesh, which is the number of holes in the mesh per inch. (See Revised Manuscript, Page 10)
Comment 25:L302: I cannot find any information about the AFT spiking protocol, nor why the particular spiked concentration was selected. Please add this information or make it more clearly identifiable.
Reply to Comment 25:Thanks for your careful comments! In the analytical performance, the linear range of the proposed method is 0.1 to 100 μg L-1, thus the intermediate value of 1μg L-1 is chosen as the spiked concentration.
Comment 26:L327: The Conclusions would benefit from further elaboration on practical implications and visions for operationalization. Could this methods be scaled up? Would it be affordable to use in practice, and for which stakeholders? How does the performance x cost trade-off compare to other methods? Cite relevant literature.
Reply to Comment 26:Thanks for your comments! We have added the corresponding discussions about the value of method actually in conclusion. (See Revised Manuscript, Page 10)
Reviewer 2 Report
The Author/Authors submitted an Article on remarkable adsorption capacity of nanomaterial as adsorbents for enrichment and detection of Aflatoxins. The Manuscript is well structured, is written according to the scientific method and it is accompanied by a rich bibliography.
However, in order to improve this precious Article, please take your precious time to review/integrate the following aspects:
Line 33: “AFT”: AFT is mentioned for the first time and, therefore, it should be indicated in full.
Line 36: Are you sure that this statement, which refers to the IARC, comes from citation no. 4? It is not possible for me to verify it since the article cited is not available on Pubmed but, above all, it seems to be related to a Chinese editor and I kindly ask you to prefer articles written in English, in order to allow referees a meticulous check.
L 38-39: The citation refers to “rice” only, are you sure you can get this type of mention to all types of “grains”? Please check it carefully.
L 52: “and so on”: please better define.
L 57: please precisely detail what is meant by "complex food matrix".
L 67-68: “to quantify trace AFT”: please review the syntax in English language.
L 259: Can I ask you if there are any specific reasons why Material and Methods chapter was later reported to Results and Discussions?
L 296: “Four corn samples, including corn, rice, soybean and millet”: Did you mean four grains samples?
LL 340-335: citations 5 and 12 are the same, please revise it.
Furthermore, to make the Article more appealing to the reader interested in Public Health topic, I would suggest adding a paragraph concerning the foods in which AFTs are known to develop, the ways to prevent their development and the limits applicable to the main food matrices, both in the Country where the study was conducted and in the European Union, given the importance of commercial transits and the free movement of goods.
Thank you for the efforts you will make to perfect this Article.
Author Response
Reply to Reviewer2
Comment 1:Line 33: “AFT”: AFT is mentioned for the first time and, therefore, it should be indicated in full.
Reply to Comment 1:Thanks for your careful comments! We have added the full form of AFT (Aflatoxin). (See Revised Manuscript, Page 1)
Comment 2:Line 36: Are you sure that this statement, which refers to the IARC, comes from citation no. 4? It is not possible for me to verify it since the article cited is not available on Pubmed but, above all, it seems to be related to a Chinese editor and I kindly ask you to prefer articles written in English, in order to allow referees a meticulous check.
Reply to Comment 2:Thanks for your careful comments! We have checked the statement, and the corresponding description has been deleted to avoid any misunderstandings. (See Revised Manuscript, Page 1)
Comment 3:L 38-39: The citation refers to “rice” only, are you sure you can get this type of mention to all types of “grains”? Please check it carefully.
Reply to Comment 3:Thanks for your careful comments! We have added the reference 5 to prove the viewpoint. (See Revised Manuscript, Page 1)
Comment 4:L 52: “and so on”: please better define.
Reply to Comment 4:Thanks for your careful comments! We have deleted the “and so on”.(See Revised Manuscript, Page 1)
Comment 5:L 57: please precisely detail what is meant by "complex food matrix".
Reply to Comment 5:Thanks for your comments! In practical applications, complex sample matrix usually refer to the electrochemically active components (e.g., catechins, gallic acid, vitamins, etc.) and inactive components (e.g., proteins, lipids, polysaccharides, etc.) present in complex food substrates that would produce strong interference and contamination signals.(See Revised Manuscript, Page 2)
Comment 6:L 67-68: “to quantify trace AFT”: please review the syntax in English language.
Reply to Comment 6:Thanks for your careful comments! We have revised the “to quantify trace AFT” to “to determine the trace AFT”.(See Revised Manuscript, Page 2)
Comment 7:L 259: Can I ask you if there are any specific reasons why Material and Methods chapter was later reported to Results and Discussions?
Reply to Comment 7:Thanks for your careful comments! We reconfirmed that this is the format required by the Journal.
Comment 8:L 296: “Four corn samples, including corn, rice, soybean and millet”: Did you mean four grains samples?
Reply to Comment 8:Thanks for your careful comments! The sentence has been revised. (See Revised Manuscript, Page 9)
Comment 9:LL 340-335: citations 5 and 12 are the same, please revise it.
Reply to Comment 9:Thanks for your careful comments! We have deleted the citation 5.(See Revised Manuscript, Page 11)
Comment 10:Furthermore, to make the Article more appealing to the reader interested in Public Health topic, I would suggest adding a paragraph concerning the foods in which AFTs are known to develop, the ways to prevent their development and the limits applicable to the main food matrices, both in the Country where the study was conducted and in the European Union, given the importance of commercial transits and the free movement of goods.
Reply to Comment 10:Thanks for your suggestion! We have added a paragraph to address this concern. (See Revised Manuscript, Page 1)
Round 2
Reviewer 2 Report
Thank you for the changes that, although mentioned in the Author response, are not sufficiently described in the text.
I kindly ask you to develop a dedicated chapter, to favor specialist readers in the topic of Food Hygiene applied to Public Health, on “the foods in which AFTs are known to develop, the ways to prevent their development and the (legal) limits applicable to the main food matrices, both in the Country where the study was conducted and in the European Union, given the importance of commercial transits and the free movement of goods”, as already indicated in the first round. The single sentence you have entered in the text is not enough. Describe the above in a detailed and in-depth manner, given the importance of both the topic debated and the journal chosen. This is a very important aspect for an article in which this topic is developed.
Furthermore, 39 citations are indicated in the article, while in the references there are 38 citations.
Author Response
Comment 1: Recently, the major hazards to human health from foods are mycotoxins, drug residues, food pathogens, heavy metal ions, food additives, and allergens. Among these, mycotoxins have attracted more and more attention from public due to its significant toxicity to humans and easy-produced in food matric. Countries around the world now are putting great effort on preventing mycotoxin development in foods to ensure the commercial transits and the free movement of goods. Reply to Comment 1: Thanks for your comments! We have added a paragraph related to the topic you mentioned. (See Revised Manuscript, Page 1) Comment 2: Furthermore, 39 citations are indicated in the article, while in the references there are 38 citations. Reply to Comment 2: Thanks for your careful comments! We have double-checked and revised the references. (See Revised Manuscript, Page 3)